

# Rapid, semi-automatic fracture and contact mapping for point clouds, images and geophysical data

Samuel T. Thiele[1], Lachlan Grose[1], Anindita Samsu[1], Steven Micklethwaite[1], Stefan A. Vollgger[1], Alexander R. Cruden[1]

[1]School of Earth, Atmosphere and Environment, Monash University, Melbourne, 3800, Australia

*Correspondence to*: Samuel T. Thiele (sam.thiele@monash.edu)

**Abstract**. Two centuries ago William Smith produced the first geological map of England and Wales, an achievement that underlined the importance of mapping geological contacts and structures as perhaps the most fundamental skill set in earth science. The advent of large digital datasets from unmanned aerial vehicle (UAV)

and satellite platforms now challenges our ability to extract information across multiple scales in a timely manner, often meaning that the full value of the data is not realised. Here we adapt a least-cost-path solver and specially tailored cost-functions to rapidly extract and measure structural features from point cloud and raster datasets. We implement the method in the geographic information system QGIS and the point cloud and mesh processing software CloudCompare. Using these implementations, the method can be applied to a variety of three-

dimensional (3D) and two-dimensional (2D) datasets including high-resolution aerial imagery, virtual outcrop models, digital elevation models (DEMs) and geophysical grids.

We demonstrate the algorithm with four diverse applications, where we extract: (1) joint and contact patterns in high-resolution orthophotographs; (2) fracture patterns in a dense 3D point cloud; (3) earthquake surface ruptures of the Greendale Fault associated with the $M_w$7.1 Darfield earthquake (New Zealand) from high-resolution light

detection and ranging (LiDAR) data, and; (4) oceanic fracture zones from bathymetric data of the North Atlantic. The approach improves the objectivity and consistency of the interpretation process while retaining expert-guidance, and achieves significant improvements (35-65%) in digitisation time compared to traditional methods. Furthermore, it opens up new possibilities for data synthesis and can quantify the agreement between datasets and an interpretation.

## 25   1   Introduction

Remote sensing datasets are commonly used in the earth sciences to interpret the morphology, location, timing and orientation of geological features. These data types, now routinely delivered by satellite, aerial and UAV platforms, have advanced to the point where they are widely available at high-resolution, and in some instances frequently updated. This proliferation of data has led to a situation where it is now no longer practical to use

manual methods to extract geological information, meaning that the full geological value of high-quality datasets is often not extracted.

For example, high and ultra-high resolution (cm to mm) photorealistic reconstructions of geological outcrops ("virtual outcrop models") are becoming widely available (Bemis et al., 2014; De Paor, 2016), typically acquired using either laser scanning technology (cf. Buckley et al., 2008) or photogrammetric workflows (cf. Bemis et al.,

2014). It is feasible to use these techniques to capture areas of several square kilometres at mm-to-cm resolution



using off-the-shelf and easy to UAV technology (e.g., Vollgger and Cruden, 2016; Cruden et al., 2016), providing for the first time an objective method for rapidly collecting detailed 3D information on geological structures.

There has recently been significant effort to develop automatic or computer-assisted methods for digitising structural data, in particular from orthorectified photographs or image sequences (Seers and Hodgetts, 2016; Vasuki et al., 2014; Jones et al., 2009). Achieving satisfactory automated digitisation is challenging for the mapping of geological structures due to intrinsic variables such as geometry, soft-linkage and segmentation over multiple scales, as well as extrinsic variables such as natural variations in colour, shadows, glare, and/or incomplete geological exposure. Due to this complexity, fully automatic methods often require significant manual adjustment and vetting to remove false positives while retaining real geological features (Vasuki et al., 2014; Seers and Hodgetts, 2016).

In this paper, we first review existing approaches to the mapping of geological structures and contacts from digital data, and then describe a novel least-cost path method that can "follow" structure traces and lithological contacts between user-defined control points in both 2D gridded datasets (photographs, geophysical imagery etc.) and dense 3D point clouds (virtual outcrop models). We then describe its implementation in two widely used software packages (QGIS and CloudCompare), and introduce four applications demonstrating the efficacy of the method for mapping outcrops, earthquake surface ruptures and oceanic fracture zones.

## 2 Existing methods

Many automated methods have been developed to extract linear features in the geosciences (e.g., Tzong-Dar and Lee, 2007; Jinfei and Howarth, 1990). These use computer vision algorithms for edge and lineament detection and, while often successful in ideal situations, require substantial fine-tuning to achieve optimal performance on real-world data. They also have a tendency to detect many false positives related to non-geological features such as shadows, roads or vegetation (Vasuki et al., 2014). Hence, even fully automated methods currently require significant manual effort to remove non-geological features while ensuring features of interest are correctly detected.

To circumvent these difficulties, several methods have been developed which remain user-driven but also use computational power to optimise the interpretation process and improve objectivity and consistency. Vasuki et al. (2014), for example, use an edge detection algorithm (phase congruency; cf. Kovesi, 1999) on orthophotographs to optimise manually defined fracture traces and contacts. This allows the user to quickly define the approximate locations of interesting features and then automatically refine them, speeding up the digitisation process significantly while avoiding problems associated with false positives. Similar computer-assisted approaches have also been applied to improve the interpretation of faults in regional magnetic surveys (Holden et al., 2016) and oceanic fracture zones in global gravity datasets (Wessel et al., 2015).

In many situations, the 3D orientations of detected features are of interest. This is typically calculated using a digital elevation model to add height information to each trace and then computing a best-fit plane (e.g., Dering et al., 2016; Jaboyedoff et al., 2009; Banerjee and Mitra, 2005). While this method works well in simple topography, it is inherently limited to 2.5 dimensions (2.5D), causing problems when features crosscut steep or overhanging topography (Pavlis and Mason, 2017). For this reason, direct analysis of 3D point cloud data is





preferable over methods that are limited to 2.5D. Unfortunately, the unstructured nature of 3D point data mean that methods for trace detection in raster data, such as those described above, cannot be easily applied.

A number of automatic methods for analysing point-cloud data have been proposed. These use clustering or plane-fitting algorithms to automatically segment and extract joint or bedding faces (*facets*) exposed on the surface of the outcrop, with reasonable success (e.g., Dewez et al., 2016; Lato and Vöge, 2012; García-Sellés et al., 2011). However, structural surfaces are not always directly exposed, and instead intersect the outcrop surface to form linear features referred to as *structural traces*. These traces cannot be detected using facet-based techniques, and require a different approach.

Seers and Hodgetts (2016) demonstrate one such approach, automatically extracting 3D structural traces by applying image-based edge detection techniques (phase congruency) to a set of images and then projecting the identified traces into 3D, using depth information derived from photogrammetric reconstructions or associated laser scan data. This approach uses multiple images to overcome issues associated with out-of-plane geometry, however as with other fully automated methods, a variety of parameters and thresholds require careful calibration and the results must be manually vetted to remove false positives.

## 3 Method: A least-cost path approach to digital mapping

### 3.1. Theory

The approach presented here couples algorithms for solving least-cost path problems with both general and use-case specific cost functions to capture structural features in both point-cloud and raster datasets. Least-cost path algorithms have previously been used to detect linear features in a variety of image data and have proven robust even when signal-to-noise ratios are very low (Sun and Pallottino, 2003; Vincent, 1998; Buckley and Yang, 1997).

Conceptually the algorithm can be divided into two steps, although for performance reasons our implementation performs these simultaneously. In the first step, data points (points in a point cloud or pixels in an image) are linked with their nearest neighbours, using a spherical search radius slightly larger than the dataset resolution, to produce a neighbourhood network (Fig. 1a). The costs of moving along links in this network (hereafter referred to as "edges") are then calculated, using a cost function designed to promote movement along structural or lithological traces and inhibit movement across them (Fig. 1b).

In the second step, an optimised version of Dijkstra's algorithm (Dijkstra, 1959) is used to derive the least-cost path between user-defined control points, providing the estimated trace (Fig. 1c). Djikstra's algorithm, in essence, progressively "grows" least-cost paths from the start point until the end is found. We optimise this by requiring paths to move closer to the end at each step, eliminating tortuous geometries that tend not to be geologically feasible. Once a trace has been estimated, manual adjustments can be easily applied by adding intermediate waypoints and recalculating the relevant least-cost paths.

The critical component in this approach is the cost function. A well-designed cost function produces low values for edges following structure or contact traces, and high values for edges outside or crosscutting traces. Our optimised implementation of Dijkstra's algorithm then follows edges with the lowest cost-values in order to map out the feature of interest. We have designed and implemented five simple cost functions that give reasonable results for different structure and data types (Appendix 1). Conveniently, simple cost functions such as point/pixel brightness or local colour gradient work well on most geological datasets; the examples presented below all map





a single scalar attribute in the dataset directly to cost (point/pixel brightness for Study 1 and 2, topographic slope for Study 3 and bathymetric depth and vertical gravity gradient for Study 4).

### 3.2. Implementation

The above methodology has been implemented as plugins for Cloud Compare (Girardeau-Montaut, 2015) and
QGIS (QGis, 2011), both of which are cross-platform, open-source and widely used software packages for geospatial analysis. Our CloudCompare plugin (*Compass*) works on point clouds, while the QGIS implementation (*GeoTrace*) works on raster data. *Compass* is bundled with the default CloudCompare distribution (since version 2.9), and the source code is freely available at https://github.com/CloudCompare/CloudCompare. Similarly, *GeoTrace* can be found on the QGIS plugin repository (https://plugins.qgis.org/plugins/), and the source code
downloaded from https://github.com/lachlangrose/GeoTrace. Complete documentation for the plugins is found at the CloudCompare wiki (http://www.cloudcompare.org/doc/wiki/) and on the *GeoTrace* GitHub page.

In addition to our method for rapidly extracting structural traces, a variety of other functionality has been implemented, including tools for measuring surface orientations, lineations and true thicknesses in the Cloud Compare plugin, and DEM based plane fitting and orientation analysis in the QGIS plugin.

**4   Case Studies**

To demonstrate the capability of the computer-assisted trace detection approach described above, we present the results of four case studies. These studies highlight the versatility of our method and its increased efficiency as compared to established manual methods.

The first case study involves the interpretation of joint sets in two 10 × 10 m areas from a ~1 cm resolution
orthophotograph of a wavecut platform at Bingie Bingie Point, New South Wales, Australia. The outcrop contains several Cretaceous to Paleogene dykes intruding Devonian plutonic diorites and tonalities and crosscut by a series of complex joint sets. The orthophotograph was generated by applying a Structure from Motion-Multi-View Stereo (SfM-MVS) workflow (Cruden et al., 2016) to digital photographs captured from a DJI S800 Evo multi-rotor UAV fitted with a 24.3-megapixel Sony Nex-7 camera and 16mm F2.8 prime lens. The two 10 × 10 m areas
(Fig. 2a, b) were selected from the survey as they contain well exposed dykes and joint sets as well as common confounding effects such as shadows and puddles. For demonstration purposes the selected areas are relatively small, but the workflow is equally applicable to much larger outcrops.

Our second case study focuses on the extraction of 3D joint traces and orientations, which are interpreted directly on a dense 3D point cloud. The Cape Woolamai sea stacks, located approximately 115 km southeast of Melbourne
on Phillip Island, have formed by erosion of the coarse-to medium-grained Cape Woolamai granite, which intruded Silurian to Lower Devonian meta-turbidites during or slightly after the mid-Devonian Tabberabberan Orogeny (a widespread episode of deformation and plutonism across Victoria; Gray, 1997; Richards and Singleton, 1981). Several sets of systematic and non-systematic joints crosscut this granite, likely related to the cooling of the intrusion, subsequent deformation and recent unloading.

For this study, a DJI Inspire 1 multi-rotor UAV and a Zenmuse X3 camera were used to capture aerial photographs, which were subsequently processed using a SfM-MVS workflow. The resulting 3D model is 45 × 40 × 25 m in size and comprises ~2 million points, with a ground sampling distance of ~2.5 cm/pixel. The topographic



complexity of this outcrop allows for accurate orientation measurement, but makes interpretation from 2.5D datasets (orthophotograph + DEM) impractical (Fig. 3a, b).

For the third case study, surface ruptures that formed along the Greendale Fault after the 2010 $M_w$7.1 Darfield earthquake are extracted from a 1 m resolution LiDAR derived DEM. The data was collected a few days after the

earthquake and was used, along with a variety of other data, to measure the surface displacement resulting from the earthquake and to interpret the kinematics of the Greendale Fault (Duffy et al., 2012).

Finally, for our last case study we interpret oceanic fracture zones in the North Atlantic from 30 arc-second bathymetry (Weatherall et al., 2015) and vertical gravity gradient data (Sandwell et al., 2014). From their inception at mid-ocean ridges, fracture zones can be used to constrain plate motion vectors and are widely used in tectonic

reconstruction (Williams et al., 2016; Sandwell et al., 2014). Both these datasets provide an opportunity to test our method on global-scale geophysical data.

## 5    Results

In each of the case studies described above the data have been interpreted twice, once using our computer-assisted method and once using manual workflows in QGIS (Study 1, 3 and 4) or CloudCompare (Study 2). The time

required to extract comparable amounts of structural data was significantly reduced using the computer-assisted method (Table 1). This efficiency increase was especially pronounced (61%) for the point cloud example, as manual methods for digitising linear features on 3D point clouds are particularly time consuming.

The following four subsections compare and contrast the results of both manual and assisted interpretations in more detail.

**5.1. Bingie Bingie Point**

Both areas (Fig. 2a, b) of the Bingie Bingie Point orthophotographs contain joints over a range of scales and in a variety of host rocks, as well as features that make automated interpretation challenging such as water, shadows and debris-filled joints. Fracture and contact traces were digitised manually in QGIS (Fig. 2c, d), and with the *GeoTrace* implementation of our assisted method (Fig. 2e, f). For the assisted interpretation, different cost

functions were used to pick the fractures and the dyke contacts. Fractures in the orthophotographs are clearly darker than their surroundings, so a greyscale version of the orthophotograph (easily calculated using *GeoTrace*) was used to define the shortest-path cost function during fracture digitisation. Dyke contacts were mapped using a cost function derived from the inverse of the local brightness gradient (high gradient = low cost). This was achieved by applying a Sobel filter (essentially a local gradient operator) to the previously mentioned greyscale

image, using scikit-image functionality (van der Walt et al., 2014) integrated into *GeoTrace*.

The results are visually similar to the manually derived reference interpretation (Fig. 2c-f). Closest-point differences between the manual and assisted interpretations show that the majority of traces (78% in Area 1 and 70% in Area 2) match to within 2 pixels (≈2 cm), smaller than the ambiguity of the dataset.

**5.2. Cape Woolamai**

Joints in the Cape Woolamai virtual outcrop model were interpreted in 3D using CloudCompare, first with the manual "draw polyline" tool and then using the *Compass* implementation of our method. The complex topography





of the sea-stacks makes 2.5D analysis inappropriate (Fig. 3a, b). As in the previous example, cost was defined by point brightness as fractures are defined by their darker colour.

In total, 146 joint traces were interpreted manually over ~3 hours, while 114 joint traces were digitised using the *Compass* plugin in less than an hour (Table 1). Joint orientations were estimated by calculating the least-squares

plane-of-best-fit for each trace. The ratio between the second and third eigenvectors of each trace was then used to reject arbitrary planes resulting from sub-linear traces, using a planarity threshold of 0.75 (cf. Thiele et al., 2015 for a more detailed description of this method). *Compass* does this in real-time during the digitisation processes, while orientation estimates from the manually digitised dataset were calculated as a post-processing step. The manual and computer-assisted methods resulted in 133 and 91 orientation estimates respectively.

Both sets of interpreted traces and associated orientation estimates appear to be broadly consistent for each method (Fig. 3c-f). Significantly, orientation estimates from the computer assisted method form more-pronounced clusters than equivalents estimated using the manually digitised traces. Although far from conclusive, this indicates that the computer assisted approach improves the consistency and precision of the orientation estimates.

### 5.3. Greendale Fault

Surface ruptures of the Greendale Fault form a series of en échelon fault-scarps visible in the LiDAR dataset (Fig. 4a). Our shortest-path method can be used to pick the fault scarps using a cost function where slope maps inversely with cost. This was achieved by calculating a slope raster using the QGIS DEM (Terrain models) tool and inverting it using *GeoTrace*.

As in the previous examples the assisted interpretation achieved very similar results to a manual interpretation, in

about half the time. Closest-point difference calculations between the manual and assisted traces also show the two sets of interpretation are consistently within ~1–2 pixels (~2 m).

### 5.4. Oceanic Fracture Zones

Oceanic fracture zones in the North Atlantic were digitised in *GeoTrace* using bathymetric depth to define trace cost. Comparison with an interpretation that was digitised manually shows similar accuracies to the previous case

studies, with the majority of traces within ~2 pixels, as well as an improvement of 36% in per-trace digitisation time (Fig. 5).

Additionally, we used the start and end points of oceanic fracture zones described in Matthews et al. (2011), which are based on a 2009 gravity gradient compilation (Sandwell and Smith, 2009), to constrain an otherwise unguided *GeoTrace* interpretation of an updated vertical gravity gradient dataset (Sandwell et al., 2014). This was achieved

using the vertical gravity gradient directly as the cost function, such that traces follow areas of low vertical gradient, and then solving the shortest-path between the Matthews et al. (2011) start and end points.

The results (Fig. 5d) again highlight the tool's general accuracy, with 65% of traces falling within 2 pixels of the Matthews et al. (2011) interpretation and 79% within 5 pixels. Most errors occurred in areas of closely spaced fracture zones, where the computed shortest-path for many fracture zones would "detour" through adjacent low-

cost features (Fig. 6). A small number of additional control points along these traces resolve this issue (Fig. 6c) by forcing the computed path to stay in the local cost-minima (the correct fracture zone), rather than taking advantage of larger adjacent minima.





## 6. Discussion

The four case studies presented above highlight applications of the least-cost-path method to the interpretation of high-resolution aerial orthophotographs, 3D point clouds, LiDAR DEM and bathymetric data - all datasets commonly used in the earth sciences to interpret and characterise geological features. We discuss here three aspects of our least-cost-path approach: it improves objectivity and reproducibility, allows automatic refinement if better data becomes available, and unlike fully-automated workflows, operates in co-operation with expert guidance.

Firstly, the approach is more objective than manual digitisation. Although not as objective as fully-automated methods (the location of the trace start and end are interpreted), most of the length of each trace is determined algorithmically, and hence will consistently locate in the same spot. Indeed, as demonstrated in Figure 7, the calculated shortest path varies only slightly when control points are interpreted at different locations. Results from Study 2 indicate that this improved consistency might increase the precision of the derived orientation estimates (Fig. 3e, f). Furthermore, each control point can easily be stored, providing a record of the locations at which interpretive decisions were made.

Similarly to the method outlined by Wessel et al. (2015) for extracting oceanic fracture zones, these control points can also be reused to generate an updated interpretation if higher-resolution or more accurate information become available. This possibility is demonstrated in Study 4, where published oceanic fracture zones were reconstructed automatically using an updated underlying dataset and the start and end points of a previous interpretation (Matthews et al., 2011). Although some quality control is required after such an operation, the digitisation process no longer needs to be completely repeated, and interpretations can be rapidly updated as datasets evolve.

Where multiple datasets are available, the similarity and total cost of paths reconstructed using different datasets can be used to quantitatively assess the degree to which different datasets support an interpretation. It is common in the geosciences to bring interpretations from multiple types of data into a single synthesis (e.g., Seton et al., 2016; Blaikie et al., 2017), especially when using geophysical datasets such as gravity and magnetics. Limiting factors during such data synthesis include both the time required and highly subjective nature of multi-data type interpretations, so a method for rapidly quantifying the extent to which different datasets support an interpretation serves as an important addition. Similarly, sensitivity analyses could be performed by randomly moving control points and measuring the response of the traces to quantify the robustness of the interpretation to uncertainty (similar to Fig. 7).

The time it takes for users to manually interpret datasets using *GeoTrace* or *Compass* will vary significantly between users, and the purpose of this study was not to comprehensively measure the efficiency of our approach. Nevertheless, in each of the case studies, our initial assessment indicates that computer-assisted interpretation required ~35-66% less user effort, as measured by both average time and mouse-clicks per structure trace, when compared to manual methods (Table 1). The resulting traces also appear to be comparable to manual traces in each case (~±2 pixels), demonstrating that our method can be used to achieve equivalent results.

The *Compass* implementation of the technique produces especially impressive results, reducing interpretation time in the Cape Woolamai example by 61%. This is pertinent given the rapid growth in both size and availability of high-resolution point cloud data and the limited range of available tools for extracting structural data from them. Significantly, the implementation of our least-cost-path method in *Compass* requires only local information,



such that calculation time scales with trace length and not dataset size. This means the tool can be used to interpret arbitrarily large point clouds.

Finally, the computer-assisted philosophy behind our method keeps the expert in control of the entire digitisation process, allowing data vetting and correction during digitisation. The approach ensures the expert becomes

familiar with the particular intricacies of each dataset, a key part of further data analysis and something not possible using automated methods yet essential for the creative process of understanding and interpreting spatial information.

## 7. Conclusions

We have described a least-cost-path based method for the computer-assisted digitisation of structural traces in

point cloud, image and raster datasets. The method enhances an expert's ability to extract geological information from the wide range of high-resolution data available to geoscientists while reducing the required time and effort. In summary, the method:

- Allows expert-guided interpretation in a way that seamlessly utilises computing power to significantly optimise the interpretation process and improve objectivity and consistency.

- Can be applied to both raster and point-cloud datasets. This is particularly significant in situations where complex topography prevents a more conventional 2.5D raster based workflow.

- Requires only local knowledge of a dataset, so that the total dataset size does not affect performance; thereby allowing computer-assisted interpretation of exceedingly large datasets.

- Is implemented as two freely available and open-source plugins for the widely used CloudCompare and

QGIS software packages.

### Data Availability

Datasets used for the Bingie Bingie Point and Cape Woolamai case studies are freely available from https://doi.org/10.4225/03/5981b31091af9. The bathymetric and vertical gravity gradient datasets used for the oceanic fracture zone example can be downloaded from the University of California San Diego at

http://topex.ucsd.edu/grav_outreach/, while the Greendale Fault LiDAR dataset is available on request from the authors of Duffy et al. (2012).

### Author Contributions

ST and LG developed the methodology described in this study. ST implemented it in CloudCompare and LG implemented it in QGIS. All of the authors contributed to the case studies and helped prepare the manuscript. The

authors declare that they have no conflict of interest.

### Acknowledgements

The authors would like to gratefully acknowledge Daniel Girardeau-Montaut and other CloudCompare developers for creating a fantastic software package and for their assistance creating the *Compass* plugin. ST was supported



by a Westpac Future Leaders Scholarship and Australian Postgraduate Award. LG was supported by an Australian Postgraduate Award. AS was supported by a Monash University Faculty of Science Dean's International Postgraduate Research Scholarship and an American Association of Petroleum Geologists Grants-in-Aid award.

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



**Appendix 1**

We outline five simple cost functions that give reasonable results for different structure types. Each function is designed to give values between 0 and 1, allowing combinations of functions to be used (by summation), and to work on both unstructured datasets (i.e. point clouds) and structured datasets (images). Hence, we do not present

any cost functions that rely on commonly used image processing techniques such as edge enhancement, although these could be easily incorporated for raster datasets. These functions are implemented directly in the *Compass* plugin, while simple QGIS functionality can be used to apply them to raster data for use with *GeoTrace*.

**Colour Brightness**

The brightness of an edge's end colour ($e_{RGB}$) can be mapped directly to edge cost (the brightness of an edges start

colour will be incorporated into the previous edge in the path). Despite its simplicity, this function (Eq. 1) is surprisingly effective at picking fracture traces, which are typically darker than their surroundings due to shadowing. Similarly, bright traces such as thin quartz or calcite veins can be identified using the opposite of this cost function (Eq. 2). Note that the division by 3 ensures that the function maps to the 0 – 1 range (assuming red, green and blue values also range from 0 to 1).

$$\mathrm{cost} = \frac{e\_R + e\_G + e\_B}{3} \qquad\qquad (Eq.\ 1)$$

$$\mathrm{cost} = 1 - \frac{e\_R + e\_G + e\_B}{3} \qquad\qquad (Eq.\ 2)$$

**Colour Similarity**

A similar cost function, based on colour similarity rather than brightness alone, is useful in more generic situations where traces have a distinctive colour but are not necessarily darker or lighter than their surroundings. This function (Eq. 3) considers an edge to be low cost if: (1) the start and end colours are similar, and; (2) the start and end colours are similar to the colour of the start and end of the trace ($B_{RGB}$ and $E_{RGB}$), minimizing along-path gradient and maximizing similarity with the trace start and end points. This function works well when traces have

a specific colour, such as for cemented joints, though it is comparatively slow compared to the brightness-based functions described above due to the large number of square roots. Similarly to the previous equations, the factors of $\sqrt{3}$ ensure that the function maps to the 0 – 1 range.

$$\mathrm{cost} = \frac{1}{2} \times \frac{|s_{RGB} - e_{RGB}|}{\sqrt{3}} + \frac{1}{2} \times \frac{|s_{RGB} - B_{RGB}| + |s_{RGB} - E_{RGB}| + |e_{RGB} - B_{RGB}| + |e_{RGB} - E_{RGB}|}{4\sqrt{3}} \qquad (Eq.\ 3)$$

**Gradient**

The previous cost functions are useful for identifying discrete structural traces such as faults, joints or thin veins, but will not be sensitive to lithological contacts. Lithological contacts are typically defined by changes in colour, and hence we base a cost function around colour gradient to identify them. This function (Eq. 4) evaluates the





gradient $G[N]$ of the magnitude of the colour vectors across the start and end neighbourhoods $N_{start}$ and $N_{end}$. To calculate the gradient for point cloud data, we use a simple method that calculates the average distance-weighted point-to-point gradient for each neighbourhood. More complex methods would highlight contacts better, but at a computational cost. For raster data, we implement a Sobel filter to achieve equivalent results.

5    An upper limit ($l$) is applied to the gradient in order to maintain a cost value between 0 and 1. A reasonable value for this limit can be approximated by dividing the maximum change in colour magnitude ($\sqrt{3}$) by the average distance between data points. This cost function can also be improved by log-transforming it to increase the importance of gradients resulting from more subtle features.

$$cost = 1 - \frac{\min\left(G[N_{start}] + G[N_{end}], l\right)}{1} \qquad (Eq.\ 4)$$

**Curvature**

In some situations, resolution is high enough that structural traces, fractures in particular, appear as topographic ridges or valleys. Hence, we include a final cost function (Eq. 5) which considers points with a high curvature as low cost, allowing paths to 'follow' ridges and valleys, where C[N] calculates the mean curvature of a point or

15    pixel neighbourhood N, and $l$ is an arbitrarily large upper limit (that allows the log-curvature to scale from 0 to 1). Note that calculating the mean curvature of a neighbourhood is computationally expensive, so this cost function performs significantly slower than the previously described ones unless curvature is pre-computed.

$$cost = 1 - \frac{\min\left(\log(C[N_{end}]), l\right)}{1} \qquad (Eq.\ 5)$$



**Table 1. Manual vs computer-assisted digitisation for the different study areas. Percentage improvements are calculated by comparing the average time and mouse clicks per digitised trace. Each case study shows a clear reduction in digitisation time, especially for the 3D datasets where manual interpretation can be especially tedious.**

| Method | Digitisation time (h:min) | Number of traces | Improvement % | Number of mouse clicks | Improvement % |
|---|---|---|---|---|---|
| *Study 1: Bingie Bingie* | | | | | |
| *Area 1* | | | | | |
| Manual | 0:54 | 270 | - | 2253 | - |
| Assisted | 0:37 | 283 | 35% | 917 | 61% |
| *Area 2* | | | | | |
| Manual | 0:57 | 338 | - | 2509 | - |
| Assisted | 0:35 | 383 | 46% | 1122 | 61% |
| *Study 2: Cape Woolamai* | | | | | |
| Manual | 3:04 | 146 | - | 6026 | - |
| Assisted | 0:56 | 114 | 61% | 1703 | 64% |
| *Study 3: Greendale Fault* | | | | | |
| Manual | 0:18 | 74 | - | 1039 | - |
| Assisted | 0:07 | 93 | 51% | 282 | 66% |
| *Study 4: Oceanic Fracture Zones* | | | | | |
| Manual | 1:17 | 432 | - | 5731 | - |
| Assisted | 0:35 | 310 | 36% | 1265 | 69% |





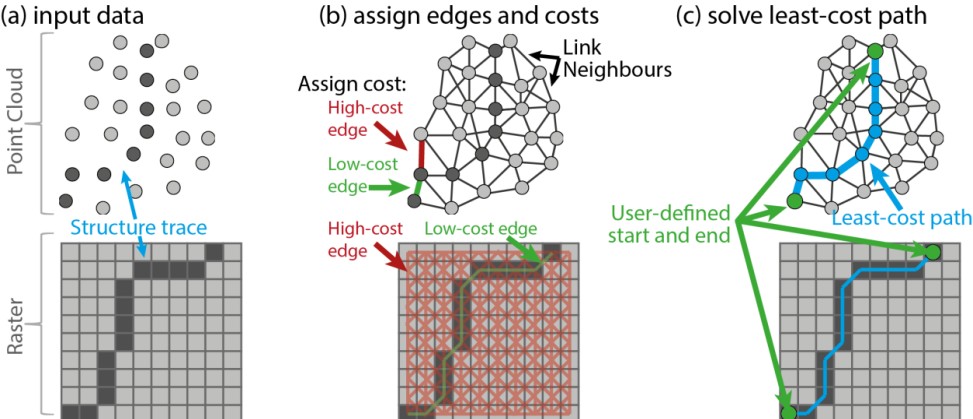

**Figure 1. Schematic representation of the least-cost path approach to trace detection for point cloud (top) and raster (bottom) data. Points/pixels on the structural trace have a lower brightness in this example (a), so a brightness-based cost function will result in low-cost edges between adjacent points/pixels that both fall on the structure trace (b). A least-cost path calculation (c) then provides an estimate of the structure trace.**

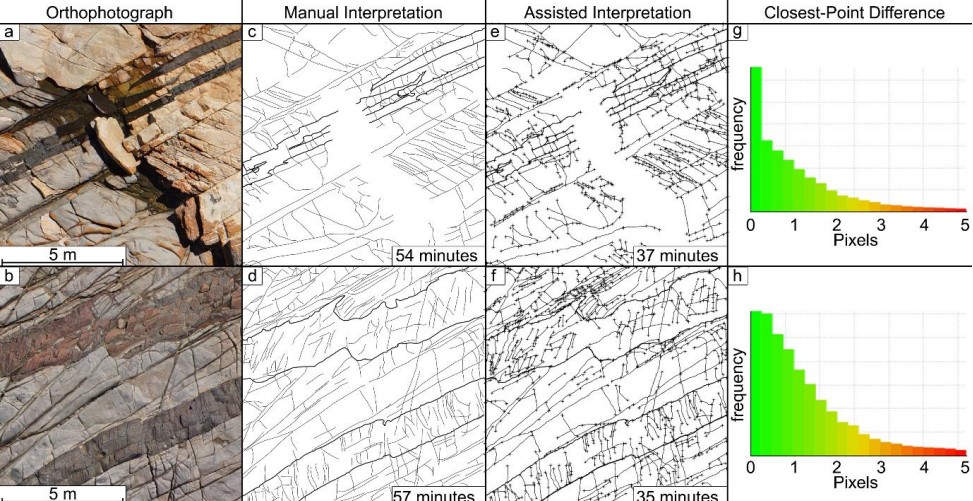

**Figure 2. The two $10 \times 10$ m orthophotographs (a, b) interpreted in Study 1. Fracture traces were digitised manually (c, d) and with our assisted method (e, f). Closest-point distances between the assisted and manual interpretations are also shown (g, h). Note the tails of these distributions have been clipped to 5 cm, as some assisted traces did not have manual equivalents, and hence gave incorrectly large closest-point differences. Small crosses in (e) and (f) represent the control points that were digitised by the user to constrain the shortest path algorithm.**





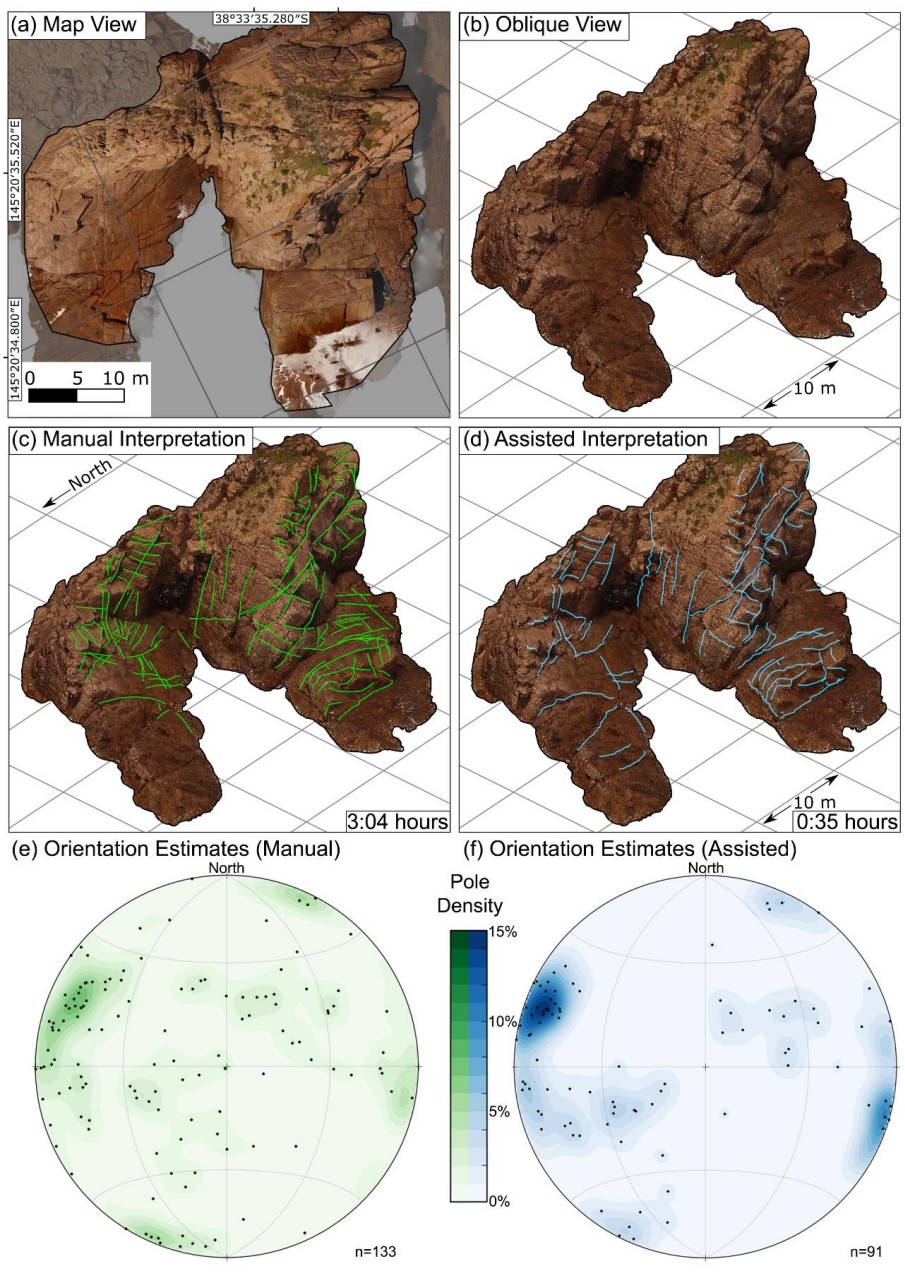

**Figure 3.** Orthophotograph of the Cape Woolamai sea stacks (a) and oblique view of the equivalent dense point cloud
(b). A 2.5D analysis conducted using the orthophotograph would significantly under sample the moderately to shallowly
dipping joint sets which are clearly visible on sub-vertical exposures in (b). Hence fractures were digitised in 3D, both
manually (c) and using the computer assisted approach (d). Equal-area lower hemisphere stereographic projections of
poles to joint orientations estimated from each of these interpretations (e-f) show that both methods produce similar
results. Poles from the computer-assisted dataset cluster more tightly (maximum density = 14.7%) than the manually
interpreted dataset (maximum density = 8.1%), indicating that the computer-assisted approach results in more
consistent orientation estimates.




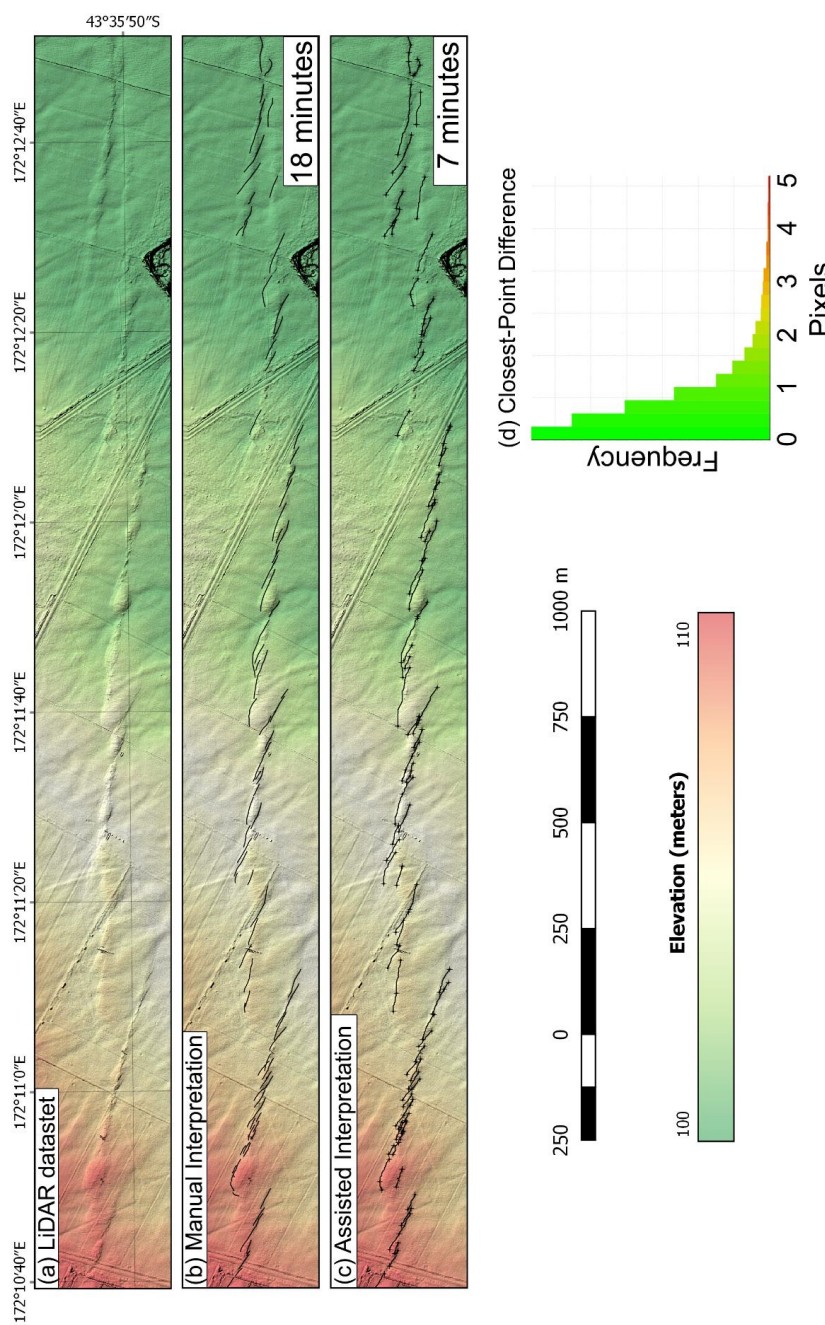

**Figure 4. LiDAR dataset illuminated from the NW showing surface ruptures of a section of the Greendale Fault, New Zealand, collected shortly after the $M_w$7.1 Darfield earthquake (a). Traces interpreted manually (b) and using the *GeoTrace* implementation of our least-cost-path method (c) are essentially equivalent (d). Control points for the assisted interpretation are shown as small crosses.**







**Figure 5.** Manual (a) and assisted (b) interpretations of oceanic fracture zones in the north Atlantic. Fracture zones interpreted manually (c) from vertical gravity gradient by Matthews et al. (2011) and reconstructed in *GeoTrace* using the start and end points only (d) are also shown. Red and blue colours in (c) and (d) show areas of high and low vertical gravity gradient, respectively. As in the previous case studies, most equivalent manual and assisted traces fall within 2 pixels (e), though differences of up to 80 pixels occur in the reconstructed dataset (d).



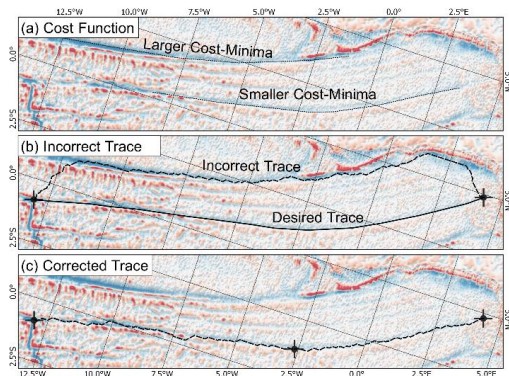

**Figure 6.** Example of a larger cost minima (a) causing the incorrect reconstruction (b) of an oceanic fracture zone. In this case the trace can be corrected by adding a single additional control point midway along the fracture zone (c).

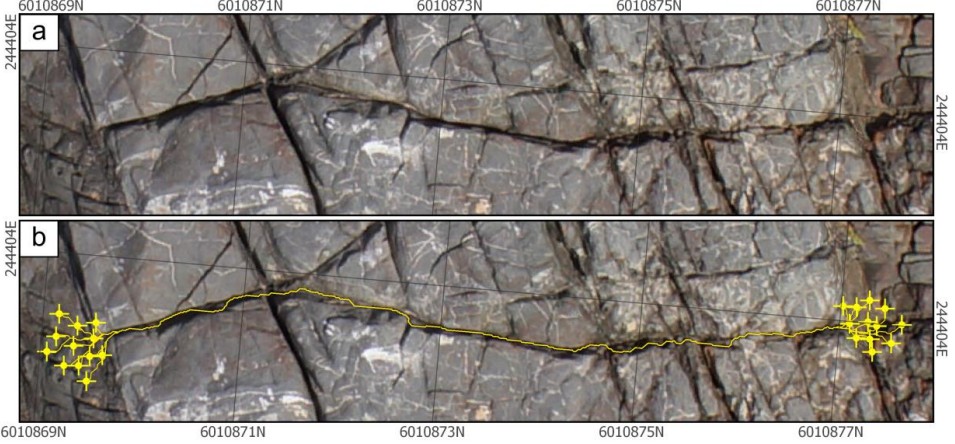

**Figure 7.** Fracture (a) from the Bingie Bingie Point dataset showing that the majority of the resulting trace (b) consistently follows the same path despite variation of the location of the control points.