# Peer review of "Rapid, semi-automatic fracture and contact mapping for point clouds, images and geophysical data"

_Solid Earth, 2017_

## Referee Comment (RC1) · A Bistacchi (Referee) · 1 Oct 2017

The manuscript se-2017-83 on "Rapid, semi-automatic fracture and contact mapping for point clouds, images and geophysical data" provides a clean explanation of an interesting algorithm that can be applied to the semi-automatic analysis of 3D point clouds from laser scanner and photogrammetry and 2D aerial/satellite images.

The manuscript is generally well written and the algorithm represents an interesting improvement with respect to usual methodologies. I have also tried to run the algorithm for 3D point clouds (distributed with CloudCompare) and the results are very good.

[Figure]

However a few points might be improved in the manuscript, hence my suggestion is for a minor review.

The most important point, in my opinion, is that the "manual" and semiautomatic interpretations yield different results (e.g. Fig 2), hence a proper discussion in terms of false positives and undetected lineaments, with a matrix showing the results, must be included. This will allow the reader to form a better idea on the value of the proposed algorithm.

Below I also list some detailed suggestions (page/line numbers in pdf):

page 1

31-32: paragraph not necessary

33: "virtual" -> consider "digital" more used nowadays

33: consider using more proper references, both older ones that introduced the digital outcrop concept and newer ones with detailed studies

34: this could be not the best reference for photogrammetric workflow

page 2

36: 2.5D should be better defined

page 3

23: add reference to Fig. 1a

24: this is Fig. 1b

27-32: some equations might help the reader here

37: even if you refer to the appendix, please list and briefly describe the cost functions here - this is a key point

page 4

4-14: have you used third party libraries or have you written all the code? this must be clearly stated here.

24: specify which software you use and how many photos in model

29: Melbourne, Australia,

35-36: specify software, camera resolution, focal length etc. (as above)

37: 2 million points is not so much with modern photogrammetry software. have you filtered the dataset? in any case please explain.

page 5

4: LIDAR-derived

29: reference for Sobel filter

31-32: explain closest-point difference - this is a key point

page 6

9: explain why you get different orientation estimates. probably a discussion in terms of false positives and undetected lineaments will be very interesting.

15-21: this case study is not described in details as the other ones. if it is not important, consider deleting it, otherwise add a description as detailed as or the others.

With best regards,

Andrea Bistacchi

---

## Referee Comment (RC2) · T. Scheiber (Referee) · 20 Oct 2017

In this article, Thiele and others present a newly-developed semi-automatic method to trace fractures and lithological contacts, which is demonstrated using a range of remote-sensed data. The article is well written, the methodological approach well-described and the case studies are presented in high-quality figures. However, fundamental concerns arose during reading the manuscript and testing the new GeoTrace plugin on a DEM. These major issues are presented in the following. I still think, though, that this is a good piece of methodological work and I can see the potential and advantage of this method which merits publication in Solid Earth provided major revisions of

the manuscript are made in line with the comments and the suggested changes below.

With best regards, Thomas Scheiber

General comments

(I) Missing description of the manual extraction method

While the semi-automatic method is well-documented and is proven to be scale-dependent, the manual method is not explained at all. There is the need for a description on how the manual mapping was performed, since it is widely known that results obtained by manual extraction are strongly dependent on various factors (cf. e.g., Scheiber et al., 2015). Especially the scale during mapping and the fractal dimension of the dataset is of great importance for this study. The larger the scale, the more details will be recognized by the mapper and, as a consequence, the more turns/curves will a structural trace have. Larger scales would thus probably lead to more similar results of the manual extraction method compared to the new semi-automatic method. In this respect, a comparison of the data by closest-point difference calculation is problematic, unless it is clearly specified under which conditions (especially the fixed scale) the manual mapping was conducted.

(II) Comparison of data

A large part of the paper, especially the result section deals with the comparison of manually-extracted data and data derived from the new semi-automatic approach. The data are compared on the basis of digitization time, number of traces, mouse-clicks (Tab. 1) and by pixel distances between constructed lines (closest-point difference, cf. Figs.2, 4,5). In the case studies the authors state that the results obtained by the two different methods are "visually similar" , "broadly consistent", "very similar" and show "similar accuracies". These phrases are way too qualitative - what is "very similar", and what would "less similar" be then? Having a closer look, however, the results of the two methods appear "not similar" to me: traces in one interpretation are longer than in

the other (i.e., they have different start and end points), there are traces drawn in one interpretation and are missing in the other, and there are traces which consist of two segments in one interpretation and appear as one trace in the other, and vice versa (see Figs. 2-5). It is not clear how the authors handled these mismatches, especially when applying a closest-point difference calculation (a description of this comparison technique is missing). The only sound comparison presented in the manuscript would be the one shown in Fig. 5f, where start- and endpoints of a previous manual interpretation are used to recalculate the structural traces using the new tool. However, in this example the authors chose to use a different (updated) base map [Sandwell and Smith (2009) in Fig. 5c versus Sandwell (2014) in Fig. 5d; p. 6, lines 27-31]. This complicates a comparison, because the resulting differences stem from both the different basemaps and the different extraction methods, and the influence of each of these variables cannot be quantified.

A proper comparison of manually- and semi-automatically produced datasets needs to fulfill the following requirements: (1) Different methods have to be tested on similar basemaps. (2) A thorough description of the fully manual method, especially by indicating the scale used while mapping is necessary. (3) The exactly same start and end point have to be used for each constructed trace. This is because the start- and end-point of the semi-automatic approach are defined manually as well! Thus it makes only sense to compare the actual traces (having the same start- to an endpoints) to figure out the differences of the fully-manual and your new method. As a consequence, the number of traces of the two opposing interpretations has to be the same. These requirements are not fulfilled in the presented case studies and should be considered when revising the manuscript. Digitization time and number mouse clicks will vary significantly between users, as correctly stated by the authors (p. 7, 30/31). So if a comparison based on these numbers is presented, it has to be clearly stated that the exercises were conducted by one and the same user.

(III) Practical application of the GeoTrace plugin

When installing the plugin (MS Windows), it would be much more user-friendly, if the installation would run automatically. Up to now users need to install and run several codes and files in python in order to make the plugin work (see detailed instructions for MS Windows users on https://github.com/lachlangrose/GeoTrace). There must be an easier, more user-friendly solution. I have tried to run the method on a .dem file and an ovr file. While it worked out well with the .dem file, I got error messages and couldn't use the plugin with the .ovr file.

"Fit planes", "Stereonet" and "Rose": I tested the tool by tracing straight and sharp bedrock lineaments, which correspond to subvertical fault and fracture zones. In case the "fit planes" box is activated, the columns called DIP and DIP_DIR in the attribute table are filled with calculated values for each trace. The values in these columns, however, appear to be incorrect: In the column DIP_DIR occur numbers which do not represent dip direction (they even include negative values), but these values appear to be arbitrary. For the dip (column DIP), I got values between 0 and 10 degrees, which is obviously wrong as well. This issue needs to be fixed and the methods used for the calculation of dip and dip direction need to be explained in the manuscript (or in the appendix). The tabs called "Stereoplot" and "Rose" allows for directly plotting the structural data. To make the tool more user-friendly, I suggest to predefine and fix the fields "Direction" and "Dip" to the columns DIP_DIR and DIP in both tabs. In both these tabs, it is not clear what the checkbox "Dip Direction" means and does. For the stereonet it needs to be indicated if the plot corresponds to lower/upper hemisphere and equal area/angle. And for the "Rose" tab, it has to be stated if the plot is number/area-weighted. In the manuscript, as it is now, no stereoplots and no rose diagrams produced by the GeoTrace tool are shown. The authors need to provide orientation data plotted in stereoplots and rose diagrams (produced by GeoTrace) for at least one case study to show the functionality and the full output spectrum of their tool.

Specific comments

page 1

Abstract

7-9: The first sentence of the abstract is far-fetched, especially when regarding the fact that mapping at this time concentrated on lithologies and lithological boundaries, not specifically structures. I therefore suggest to delete this sentence.

12: "...extract..." There is the need to add a sentence describing how your new extraction method practically works - you manually add a start and end point between which a connecting line is calculated, which can be manually tuned by adding additional points in between.

21: "The approach improves the objectivity and consistency..." Since start and end points are set manually, I don't really see how this method then improves objectivity

21: change "expert" to "user". This accounts for all places where "expert" is used in the manuscript. Not every user can be considered an expert.

23-24: "it... can quantify the agreement between datasets and interpretation." Unclear. What interpretation (manual or semi-automatic)? How is this practically done? - In the GeoTrace plugin, I couldn't find any tool to compare data - ?

Introduction

33: add e.g., to the references; refer only once to Bemis et al., 2014

35 and following: rephrase this sentence

page 2

1: something is missing between easy and to

11: remove "and contacts". Contacts are geological structures.

13: in brackets: name only the sort of data you used in your case studies.

Existing Methods

Add a short description of the fully manual method here as well and refer to its drawbacks regarding objectivity. Refer to Scheiber et al. (2015). Scheiber, T., O. Fredin, G. Viola, A. Jarna, D. Gasser, and R. Łapińska-Viola (2015):Manual extraction of bedrock lineaments from high-resolution LiDAR data: methodological bias and human perception, GFF, 137, 362-372, doi: 10.1080/11035897.2015.1085434.

16: write more specific: "outcrop structures" instead of "outcrops"

18/19: refer to more recent papers as well

33 onwards: explain the dimensions 3D versus 2.5D

35/36: what do you consider a simple topography? Be more precise.

page 3

2: refrain from using phrases such as "such as described above". Follow this suggestion throughout the manuscript.

4: use the more general term "fractures" instead of "joints"

Method

page 4

14: "DEM based plane fitting and orientation analysis" didn't work (see general comment III)

16: To demonstrate the capability of our new computer-assisted trace detection approach, ...

18: "established manual methods" - A thorough description of the manual method is of great importance as well: Who mapped? At which scale was the mapping performed (see general comment I)?

21: "plutonic diorites and tonalites" - either "plutonic rocks" or "diorites and dolerites" instead

28: use "fractures" instead of "joints" throughout

33: "Several sets of systematic and non-systematic joints..." if a joint/fracture is non-systematic, it doesn't belong to any set.

34: "cooling of the intrusion, subsequent deformation and recent unloading." Are these guesses? If not add references, otherwise remove it. What does "recent" mean in geological time scale? Be more precise.

page 5

1: "...accurate orientation measurement" - of what?

Results

14: "manual workflows" - the manual extraction method needs to be described for each case study.

18: "...compare and contrast the results of both manual and assisted interpretations..." The result sections dealing with comparison of the data have to be rephrased in line with newly obtained results (see general comment II). And the closest-point difference calculation needs to be explained in the method section.

29: remove "previously mentioned"

31-33: "The results are visually similar..." - see general comment II

page 6

1: "As in the previous example,..." Name it.

3: What does this difference in numbers of traces reflect? (see also comment to table 1 and general comment I and II)

8: "as a post processing step..." shorter: "after processing"

15: "fault scarps" In the LiDAR DEM (Fig. 4a) the interpreted fault scarps (Figs. 4b and

c) are not visible at all. The slope map which was used for doing the interpretation has to be provided in Fig. 4.

27: Replace "described in" by "interpreted by"

33 onwards: Here it is unclear whether the errors refer to the 21% pixels located >5 pixels away. "the computed shortest-path ... would "detour" through..." Does this mean that you reduced these errors by adding control points to "guide" the trace? Clarify

36 "correct fracture zone" A fracture zone cannot be correct or incorrect. Change to for example "desired structural trace"

page 7

Discussion

3: DEM: use plural: DEMs

6: "operates in co-operation" Rephrase

6/7: change to "user guidance"

8: if there is a firstly, there has to be somewhere a secondly as well

12: "improved consistency" Yes there is improved consistency, but it doesn't necessarily mean that it increases precision and that it is closer to reality. This remains a bit abstract unless a reference to the true (nature) pattern of the fractures is given.

30 "...manually interpret datasets using GeoTrace or Compass..." this is confusing. GeoTrace and Compass are semi-automatic tools. I suggest to use "manual" for the fully-manual extraction method and "semi-automatic" or "computer-assisted" for your new extraction tool consistently throughout the manuscript.

34/35 cf. general comment II

page 8

6: add "fully" to "automated"

Tables and Figures

Table 1:

Why do you get different numbers of traces for the manual and the assisted method? Is this due to different operators for each method? Or does this reflect different perception states of the same operator? What does the comparison in numbers tell us here? Cf. general comment II.

Figures 2-5 should be considered to get redrawn/rearranged according to the general comment II and the comments below.

Figure 2: North arrow is missing in both (a) and (b). Here you could add stereoplots and rosediagrams produced by GeoTrace (cf. general comment III).

Figure 3: North arrow is missing in both (a) and (b). What is the reason for the more consistent orientation estimates using the computer-assisted approach (Fig. 3f)? Is this an artefact of the computer-assisted approach? Is the larger spread in orientations in the manual-extracted dataset more realistic? What is closer to reality? See also comment to page 7, line 12. Discuss.

caption: get rid of the "similar results" (see general comment II).

Figure 4: North arrow is missing. Scale bar: show it from 0 to 1000m only, don't go to -250. What are the straight lines across the DEM? Roads or tractor tracks? Indicate. Show the map you used for extraction of structural traces. In the un-interpreted hillshaded DEM (Fig. 4a) I would not dare drawing any of the interpreted traces. Show the slope map, which was used to extract the traces.

Figure 5: Legend for 5a/b (bathymetry?) and for 5c/d (vertical gravity gradient) is missing. The original/uninterpreted data (basemap) needs to be shown for 5a/b. You did the mapping probably at larger scales? If yes, provide zoom-in maps showing

exemplary oceanic fracture zone and their interpretation. Indicate location of Fig. 6 in Fig. 5c or 5d.

caption: remove the last sentence (see general comment II)

Figure 5: Add scale and north arrow.

Technical corrections

page 6

1: "sea-stacks" - in other places written without hyphen. Be consistent.

---

## Author Comment (AC1) · 3 Nov 2017

Dear A. Bistacchi,

We thank you for your time and effort reviewing the submitted manuscript, and are pleased that you appreciated our results. We have incorporated the majority of your suggestions into the revised manuscript, as detailed in the following pages. Please note that to facilitate the evaluation of our revision, the page and line numbers of the reviewer's comments refer to the originally submitted manuscript while page and line numbers of our responses refer to our revised manuscript.

Regards,

Samuel T. Thiele, Lachlan Grose, Anindita Samsu, Steven Micklethwaite, Stefan A. Vollgger & Alexander Cruden

**Response to review by A. Bistacchi**

*General Comments:*

The most important point, in my opinion, is that the "manual" and semiautomatic interpretations yield different results (e.g. Fig 2), hence a proper discussion in terms of false positives and undetected lineaments, with a matrix showing the results, must be included. This will allow the reader to form a better idea on the value of the proposed algorithm.

Clarify: The reviewer is correct that the manual and semi-automatic interpretations yield different results. This is largely due to the inherent variability of the interpretive decision processes applied when using these types of datasets – differing results would also be expected on manually digitising the same dataset multiple times. As discussed at lines 7.28-7.34, our method improves reproducibility by rigorously and consistently "interpolating" between points of interpretation, however results still depend on these interpretive decisions, hence the "natural" variation.

Regarding false-positives and undetected lineaments, our method does not do any "lineament detection" as it relies on the user to pick each lineament (by inserting the control points). Thus, the approach is subject to identical "false-positives" and "undetected-lineaments" as manual interpretations, something that we feel is outside the scope of this paper.

Situations where the semi-automatically generated trace does not match the user's expectations (i.e. they expected the algorithm to follow a different lineament) could be considered a false positive, though the inclusion of additional constraints (intermediate "waypoints") can be used to easily modify the results to meet expectations. This is discussed in the text at lines 8.26-8.30, with a specific example provided in Fig. 6.

*Specific Comments:*

31-32: paragraph not necessary

Agree: The paragraph-break has been removed. [Line 1.31]

33: "virtual" -> consider "digital" more used nowadays

Agree: "virtual" has been replaced with "digital" throughout the text.

33: consider using more proper references, both older ones that introduced the digital outcrop concept and newer ones with detailed studies

Agree: The reference has been updated as suggested, and now includes references to McCaffrey et al. (2005), Pringle et al. (2006) and Vollgger and Cruden (2016). [Line 1.30]

34: this could be not the best reference for photogrammetric workflow

Clarify: Bemis et. al. provides a useful and widely acknowledged review of the photogrammetric workflow as applied to structural geology and seismology. Hence, we believe that it is an appropriate reference in this context. To broaden the scope somewhat we have included an additional reference to Smith et al. (2015). [Line 1.31]

**36: 2.5D should be better defined**

Agree:  The sentence has been rephrased as follows:

*"While this method works well in topography containing slopes <45°, it is inherently limited to 2.5 dimensions (2.5D) as elevation data is gridded in the X-Y plane, causing problems when features crosscut steep or overhanging topography (Pavlis and Mason, 2017)"* [Line 2.33-2.35]

**23: add reference to Fig. 1a**

Agree: The text has been modified such that each part of the figure is now explicitly referenced:

*"In the first step, data points (points in a point cloud or pixels in an image; Fig. 1a) are linked with their nearest neighbours, using a spherical search radius slightly larger than the dataset resolution, to produce a neighbourhood network (Fig. 1b). The costs of moving along links in this network (hereafter referred to as "edges") are then calculated, using a cost function designed to promote movement along structural or lithological traces and inhibit movement across them (Fig. 1c)."* [Lines 3.20-3.24]

**24: this is Fig. 1b**

Agree: This has been corrected (see above).

**27-32: some equations might help the reader here**

Clarify: Dijkstra's algorithm is a recursive path-growing procedure, and so cannot be simply expressed as an equation. Hence, we suggest that attempting to do so would only confuse the reader. Furthermore, detailed descriptions of the method and its implementations are readily available, including in the cited Dijkstra et al., 1959.

**37: even if you refer to the appendix, please list and briefly describe the cost functions here - this is a key point**

Agree: The cost functions used in this case study are now listed and described in the text:

*"… simple cost functions such as point/pixel brightness or local colour gradient work well on most geological datasets; the examples presented below all map a single scalar attribute in the dataset directly to cost (point/pixel brightness for Study 1 and 2, topographic slope for Study 3 and bathymetric depth and vertical gravity gradient for Study 4).  We have designed and implemented these and several other simple cost functions that give reasonable results for different structure and data types. Specific equations for these are included as Appendix 1."* [Lines 3.34-4.2]

For clarity and brevity we have kept the specific equations in the appendix.

**4-14: have you used third party libraries or have you written all the code? this must be clearly stated here.**

**Agree:** The Compass plugin uses no third-party libraries other than CloudCompare and its dependencies. The GeoTrace implementation uses numpy data structures and scikit-image functionality. The text has been amended to reflect by adding the following:

*"Our CloudCompare plugin (Compass) is written in C++ and works for point cloud data, while the QGIS version (GeoTrace) is implemented as a python plugin, using numpy (van der Walt et al., 2011) and scikit-image (van der Walt, 2014) to apply our method to raster data"* [Lines 4.6-4.8]

24: specify which software you use and how many photos in model

**Agree:** The text has been updated as follows: *"… applying a Structure from Motion-Multi-View Stereo (SfM-MVS) workflow, as implemented in Agisoft Photoscan v1.2.6 to 297 digital photographs …"* [Lines 4.22-4.23]

29: Melbourne, Australia,

**Agree:** The text has been updated to specify that Melbourne is in Australia:

*"The Cape Woolamai sea stacks, located approximately 115 km southeast of Melbourne (Australia) on Phillip Island, have formed by erosion of …"* [Lines 4.32-4.33]

35-36: specify software, camera resolution, focal length etc. (as above)

**Agree:** The text has been updated to include this information:

*"For this study, a DJI Inspire 1 multi-rotor UAV and 20 mm fix-focal Zenmuse X3 camera were used to capture 274 aerial photographs, which were subsequently processed using Agisoft Photoscan."* [Lines 5.1-5.2]

37: 2 million points is not so much with modern photogrammetry software. have you filtered the dataset? in any case please explain.

**Agree:** The 2 million points represents a small subset of a much larger survey, hence the relatively small number of points. To clarify the text has been modified as follows:

*"A 45 × 40 × 25 m section of the resulting model containing a single sea-stack was then extracted, containing ~2 million points and representing an average ground sampling distance of ~2.5 cm/pixel."* [Lines 5.2-5.4]

4: LIDAR-derived

**Agree:** As suggested, a hyphen has been added to the manuscript. [Line 5.8]

29: reference for Sobel filter

**Agree:** An appropriate citation to Sobel (1990) has been included [Line 6.1]

31-32: explain closest-point difference - this is a key point

**Agree:** The manuscript has been amended to describe the closest-point difference calculation:

"*Closest-point differences, calculated by subsampling closely-spaced points from each assisted trace and computing the shortest-distance between these and a manually interpreted trace, show that…*"
[Line 6.12 – 6.14]

9: explain why you get different orientation estimates. probably a discussion in terms of false positives and undetected lineaments will be very interesting.

**Agree:** The differing number of orientation estimates reflect the different number of traces digitised manually (146) and with the computer assisted method (114). The difference in the resulting stereoplots (Fig. 3.) probably results from the larger number of points in the assisted traces (which sample every point along each trace) as compared to manually digitised traces: Manually digitised traces only contain points picked by the user during digitisation, and hence will be more affected by outliers. The text has been updated to reflect this: "*Although far from conclusive, this indicates that the computer-assisted approach improves the consistency and precision of the orientation estimates, likely due to the larger number of points it samples. The assisted method samples every point along each trace, while the manual method only includes the polyline vertices created during digitisation, making the best-fit plane more susceptible to errors caused by outliers.*"  [Lines 6.30-6.34]

15-21: this case study is not described in details as the other ones. if it is not important, consider deleting it, otherwise add a description as detailed as or the others.

**Clarify:** The Greendale Fault case study demonstrates an application of our method to LiDAR data, which is commonly used throughout the geosciences. Hence we feel that it is worth retaining even if it is not discussed in detail. Furthermore, a detailed description of this dataset is not necessary as: (1) the results are quite similar to those from the Bingie Bingie case study, and hence can be communicated succinctly, and; (2) a full interpretation and description of this dataset has been previously published by Duffy et al., 2012, as mentioned in the manuscript.

Dear T. Scheiber,

We thank you for your time and effort reviewing the submitted manuscript and for your constructive suggestions. We have incorporated the majority of these into the revised manuscript, as detailed in the following pages. Please note that to facilitate the evaluation of our revision, the page and line numbers of the reviewer's comments refer to the originally submitted manuscript while page and line numbers of our responses refer to our revised manuscript.

Regards,

Samuel T. Thiele, Lachlan Grose, Anindita Samsu, Steven Micklethwaite, Stefan A. Vollgger & Alexander Cruden

General Comments

*(I)        Missing description of the manual extraction method*

While the semi-automatic method is well-documented and is proven to be scaledependent, the manual method is not explained at all. There is the need for a description on how the manual mapping was performed, since it is widely known that results obtained by manual extraction are strongly dependent on various factors (cf. e.g., Scheiber et al., 2015). Especially the scale during mapping and the fractal dimension of the dataset is of great importance for this study.

Agree: The reviewer is correct that our description of the manual digitisation method used is inadequate. This has been rectified by adding two paragraphs describing the manual methods to the end of Section 4. These are as follows:

*"For each of these case studies, we also assess the similarity of our assisted results to manually derived interpretations. The aim of these comparisons is not to rigorously validate the computer-assisted method as: (1) given enough control-points our method could match any interpretation, and; (2) manual interpretation of structures from remotely sensed data is notoriously subjective, and so differing results can be expected from different operators (Bond et al., 2007) and scales (Scheiber et al., 2015). Instead, we seek to demonstrate the applicability and versatility of the assisted approach, and that similar results to a manual interpretation can be produced for less time and effort.*

*For each interpretation, the operator was instructed to digitize every structural feature within the dataset. To ensure this was an achievable task, the extent of the dataset used in each case study is small compared to its resolution. No attempt was made to ensure that the same number of features was extracted from each dataset, as this would affect timing measurements. Digitization was performed at or close-to the dataset resolution, so that the manual interpretations contained similar detail to the assisted approach (which always follows traces at the resolution of the dataset), although the operator was able to freely zoom in and out to inspect the data at multiple scales. The same operator performed the manual and assisted interpretations for Case Studies 1 and 2, while different operators generated each of the assisted and manual interpretations for Case Study 3 and 4. Manual interpretation in QGIS (Case Study 1, 3 and 4) was performed by digitising polyline features to a shapefile using the "Add feature" tool, while in CloudCompare (Case Study 2) the "Trace-polyline by point picking" tool was used."* [Line 5.16-5.32]

The larger the scale, the more details will be recognized by the mapper and, as a consequence, the more turns/curves will a structural trace have. Larger scales would thus probably lead to more similar results of the manual extraction method compared to the new semi-automatic method. In this respect, a comparison of the data by closest-point difference calculation is problematic, unless it is clearly specified under which conditions (especially the fixed scale) the manual mapping was conducted.

Clarify: Each of case studies chosen in this paper represent small subsets of much larger datasets, and hence the interpretations were performed over a limited range of scales at or close-to the dataset resolution. The assisted-method always "follows" traces at the largest possible scale (i.e. at the data resolution), and hence will be scale-independent for individual traces (though the user-decision on what comprises a single-structure/trace will be scale dependent). Thus, we suggest that the comparison between the two (automatic vs assisted) is reasonable, especially given the limited and general conclusions we draw from these.

This has been clarified on lines 5.24– 5.28 (see response to previous point).

Finally, we are unsure why the reviewer considers that larger-scales would lead to smaller differences between the assisted and automated approaches – obvious errors in the digitisation with the assisted approach are corrected by the user during digitisation (as discussed at lines 8.26 –8.30), reducing the incidence of large errors at small scales.

*(II) Comparison of data*

A large part of the paper, especially the result section deals with the comparison of manually-extracted data and data derived from the new semi-automatic approach. The data are compared on the basis of digitization time, number of traces, mouse-clicks (Tab. 1) and by pixel distances between constructed lines (closest-point difference, cf. Figs.2, 4,5). In the case studies the authors state that the results obtained by the two different methods are "visually similar" , "broadly consistent", "very similar" and show "similar accuracies". These phrases are way too qualitative - what is "very similar", and what would "less similar" be then?

**Clarify:** The phrases are indeed qualitative, though the authors suggest that is reasonable given: (1) the inherent variability of manual data interpretation (as correctly pointed out by the reviewer, and described in lines 5.18-5.20), and; (2) our aims from the comparison are only to (qualitatively) demonstrate that results similar to a manual interpretation can be produced with the assisted-method (rather than a robust validation; now stated at lines 5.20-5.22)

Furthermore, we include both quantitative histograms of the closest-point distance between the two interpretations for each Case Study (Figs. 2, 4, and 5) and the percentage of the traces that fall within threshold distances of 2 and 5 pixels. These comparisons are entirely quantitative and are provided throughout the text (lines 6.14-6.15, 7.4-7.5, 7.9 and 7.16-7.17) as justification for the qualitative statements the reviewer has identified.

Having a closer look, however, the results of the two methods appear "not similar" to me: traces in one interpretation are longer than in the other (i.e., they have different start and end points), there are traces drawn in one interpretation and are missing in the other, and there are traces which consist of two segments in one interpretation and appear as one trace in the other, and vice versa (see Figs. 2-5). It is not clear how the authors handled these mismatches, especially when applying a closest-point difference calculation (a description of this comparison technique is missing).

**Clarify:** The reviewer is correct that the number and length of traces in each interpretation differ. This is an expression of the natural variability of the interpretation processes and, as now clearly stated at line 5.25, no attempt was made to reduce this (by, for example, enforcing traces to have the same start/end points) as doing so would significantly affect the timing comparisons.

Such variations in the dataset were not corrected for in the closest-point difference calculations as this technique does not require exactly analogous traces (as now described on lines 6.12-6.14). This is the reason the histograms in Figs. 2, 4 and 5 have such long tails. In Case Study 1, these were clipped at a large distance to remove points from traces that did not have equivalents in both datasets, as described in the captions of Fig. 2.

The only sound comparison presented in the manuscript would be the one shown in Fig. 5f, where start- and endpoints of a previous manual interpretation are used to recalculate the structural traces using the new tool. However, in this example the authors chose to use a different (updated) base map [Sandwell and Smith (2009) in Fig. 5c versus Sandwell (2014) in Fig. 5d; p. 6, lines 27-31]. This complicates a comparison, because the resulting differences stem from both the different basemaps and the different extraction methods, and the influence of each of these variables cannot be quantified. A proper comparison of manually- and semi-automatically produced datasets needs to fulfill the following requirements: (1) Different methods have to be tested on similar basemaps. (2) A thorough description of the fully manual method, especially by indicating the scale used while mapping is necessary. (3) The exactly same start and end point have to be used for each constructed trace. This is because the start- and end-point of the semi-automatic approach are defined manually as well! Thus it makes only sense to compare the actual traces (having the same start- to an endpoints) to figure out the differences of the fully-manual and your new method. As a consequence, the number of traces of the two opposing interpretations has to be the same. These requirements are not fulfilled in the presented case studies and should be considered when revising the manuscript.

**Clarify:** The method the reviewer suggests for comparing the assisted and manual approaches using fixed start/end points for each trace would produce interesting results, however this approach would make the timing/mouse-click measurements meaningless, as mentioned at line 5.25. Furthermore, one of the key advantages of our computer-assisted approach is that a user can iteratively modify/correct results in real-time (cf. lines 8.26 –8.30), making the "accuracy" of the results essentially dependent on the effort spent by the user on quality control and making such a rigorous validation largely arbitrary. Hence we suggest that such a comparison is outside the scope of this paper, which is focused on presenting our new method and demonstrating ways it can be applied.

We make this clear by stating "*The aim of these comparisons is not to rigorously validate the computer-assisted method as: (1) given enough control-points our method could match any interpretation, and; (2) manual interpretation of structures from remotely sensed data is notoriously subjective, and so differing results can be expected from different operators (Bond et al., 2007) and scales (Scheiber et al., 2015). Instead, we seek to demonstrate the applicability and versatility of the assisted approach, and that similar results to a manual interpretation can be produced for less time and effort.*" [Lines 5.17-5.22]

The purpose of Fig 5c. is to show that our method can be used to translate interpretations between datasets (as discussed at lines 7.13, 7.35-8.4), hence the use of different datasets.

Digitization time and number mouse clicks will vary significantly between users, as correctly stated by the authors (p. 7, 30/31). So if a comparison based on these numbers is presented, it has to be clearly stated that the exercises were conducted by one and the same user.

**Agree:** The text now clearly states that "*The same operator performed the manual and assisted interpretations for Case Studies 1 and 2, while different operators generated each of the assisted and manual interpretations for Case Study 3 and 4.*" [Lines 5.28-5.30].

*(III) Practical application of the GeoTrace plugin*

When installing the plugin (MS Windows), it would be much more user-friendly, if the installation would run automatically. Up to now users need to install and run several codes and files in python in

order to make the plugin work (see detailed instructions for MS Windows users on https://github.com/lachlangrose/GeoTrace). There must be an easier, more user-friendly solution.

**Clarify:** The difficulty installing the GeoTrace plugin on Windows results largely from software dependencies of the scikit-image package used, for performance reasons, to perform the shortest-path calculations. We are currently working to create an installation script/file, however unfortunately the security features of windows and the need for admin privileges make this no easy task. A feature-request has been posted on the GitHub page to record progress.

I have tried to run the method on a .dem file and an ovr file. While it worked out well with the .dem file, I got error messages and couldn't use the plugin with the .ovr file.

**Agree:** .ovr files are a "pyramid file", containing pre-computed data tiles at different scales for rapid visualisation. This causes issues in the plugin, as it requires easy access to data at full-resolution, hence the error messages. We have modified the plugin so that it throws a "friendly" error message when .ovr files are used. QGIS can easily be used to convert .ovr files to a compatible data format.

"Fit planes", "Stereonet" and "Rose": I tested the tool by tracing straight and sharp bedrock lineaments, which correspond to subvertical fault and fracture zones. In case the "fit planes" box is activated, the columns called DIP and DIP_DIR in the attribute table are filled with calculated values for each trace. The values in these columns, however, appear to be incorrect: In the column DIP_DIR occur numbers which do not represent dip direction (they even include negative values), but these values appear to be arbitrary. For the dip (column DIP), I got values between 0 and 10 degrees, which is obviously wrong as well. This issue needs to be fixed and the methods used for the calculation of dip and dip direction need to be explained in the manuscript (or in the appendix).

**Clarify:** We have tested the "fit-planes" functionality on several datasets, with reasonable results. We suggest that the "arbitrary" orientations and shallow dip values might result because the best-fit-plane is sub-parallel to topography (i.e. variation in the structure orientation is resulting in an incorrect best-fit-plane). Poor-quality fits can also be obtained from highly co-linear traces. If this issue persists we would appreciate it if the reviewer would add a bug-report to the GitHub page that includes further information on the situation in which the issue arose.

To help clarify the quality of the orientation estimates, the plugin has been modified to calculate the "planarity" metric described by Thiele et al., 2015, as well as a crude classification of fit plane quality based on this. A description of this, and the plane-fitting method used (Eigen analysis), has been added to the plugin documentation. As plane-fitting is outside the scope of this publication (which focusses on the least-cost-path method for trace rapid trace digitization) we have not included this in the manuscript.

**Agree:** The plugin no longer produces negative (i.e. anti-clockwise) dip-direction bearings.

The tabs called "Stereoplot" and "Rose" allows for directly plotting the structural data. To make the tool more user-friendly, I suggest to predefine and fix the fields "Direction" and "Dip" to the columns DIP_DIR and DIP in both tabs. In both these tabs, it is not clear what the checkbox "Dip Direction" means and does.

**Agree:** The requested features have been included in the plugin. The stereonet tab has been modified to make it clear that by checking "Dip-Direction" orientations are expected in the format Dip/Dip-Direction rather than Strike/Dip.

For the stereonet it needs to be indicated if the plot corresponds to lower/upper hemisphere and equal area/angle. And for the "Rose" tab, it has to be stated if the plot is number/area-weighted. In the manuscript, as it is now, no stereoplots and no rose diagrams produced by the GeoTrace tool are shown.

Agree: The requested information is now included on the stereonet/rose-diagram plots.

The authors need to provide orientation data plotted in stereoplots and rose diagrams (produced by GeoTrace) for at least one case study to show the functionality and the full output spectrum of their tool.

Clarify: The purpose of this publication is to present the least-cost-path method for trace digitisation. The functionality for drawing stereoplots/rose-diagrams, included for convenience in the GeoTrace plugin, is not scientifically novel and is outside the scope of this paper.

**Specific comments**

*Abstract*

7-9: The first sentence of the abstract is far-fetched, especially when regarding the fact that mapping at this time concentrated on lithologies and lithological boundaries, not specifically structures. I therefore suggest to delete this sentence.

Agree: The sentence has been removed. [Line 1.7]

12: "...extract..." There is the need to add a sentence describing how your new extraction method practically works - you manually add a start and end point between which a connecting line is calculated, which can be manually tuned by adding additional points in between.

Agree: Line 1.9 has been modified to reflect this, and now reads: "*Here we adapt a least-cost-path solver and specially tailored cost-functions to rapidly interpolate structural features between manually defined control points in point cloud and raster datasets.*" [Line 1.9-1.10]

21: "The approach improves the objectivity and consistency..." Since start and end points are set manually, I don't really see how this method then improves objectivity 21: change "expert" to "user". This accounts for all places where "expert" is used in the manuscript. Not every user can be considered an expert.

Clarify: The method improves objectivity in that a significant portion of the trace length is directly guided by the underlying dataset (e.g. Fig. 7), and so will consistently be reproduced given the same input/control points. The reviewer is correct, however, that as the user chooses the start and end-points (and can add intermediate points to "force" the trace to do almost anything), so the approach is certainly not completely objective (as discussed in detail at lines 7.25-7.34).

To avoid implying objectivity without the associated caveats included in the discussion, the sentence has been modified to:

*"The approach improves the consistency of the interpretation process while retaining expert-guidance, and achieves significant improvements (35-65%) in digitisation time compared to traditional methods"* [Line 1.19].

Also note that we retain the term "expert-guidance" as: (1) it is used elsewhere in the literature (e.g. Vasuki et al., 2014), and; (2) we consider that the "user" knows best what they want to extract from the dataset, making them an "expert" by comparison with our humble computer program.

23-24: "it... can quantify the agreement between datasets and interpretation." Unclear. What interpretation (manual or semi-automatic)? How is this practically done? - In the GeoTrace plugin, I couldn't find any tool to compare data - ?

**Clarify:** This statement relates to the method we present rather than its specific implementations (GeoTrace and Compass), and is a summary of the discussion points presented in lines 8.5 – 8.14. The authors are of the opinion that the sentence is clearly a general statement about the approach rather than a specific reference to either of the plugins, and hence it has been retained.

*Introduction*

33: add e.g., to the references; refer only once to Bemis et al., 2014

**Agree:** The references at line 1.29 have been updated as per the comment of Reviewer #1, and no longer refer to Bemis et al., 2014.

35 and following: rephrase this sentence

**Agree:** The sentence has been rephrased as follows:

*"These techniques, combined with inexpensive and easy-to-use UAV technology, now make it feasible to acquire topographic data at mm-to-cm resolution over areas of several square kilometres (e.g., Vollgger and Cruden, 2016; Cruden et al., 2016), providing for the first time an objective method for rapidly collecting detailed 3D information on geological structures."* [Line 1.31 – 1.34]

1: something is missing between easy and to

**Agree:** The missing word has been added (see above).

11: remove "and contacts". Contacts are geological structures

**Agree:** "and contacts" has been removed as suggested. [Line 2.9]

13: in brackets: name only the sort of data you used in your case studies.

**Agree:** The listed datasets now explicitly relate to data types presented in this study:

" ... 2D gridded datasets (imagery, geophysical rasters etc.) and dense 3D point clouds (digital outcrop models)." [Line 2.11]

*Existing Methods*

Add a short description of the fully manual method here as well and refer to its drawbacks regarding objectivity. Refer to Scheiber et al. (2015). Scheiber, T., O. Fredin, G. Viola, A. Jarna, D. Gasser, and R. Łapi´nska-Viola (2015):Manual extraction of bedrock lineaments from high-resolution LiDAR data: methodological bias and human perception, GFF, 137, 362-372, doi: 10.1080/11035897.2015.1085434.

Agree: Please refer to our response to General Comment I

16: write more specific: "outcrop structures" instead of "outcrops"

Agree: The sentence has been amended as suggested. [Line 2.13]

18/19: refer to more recent papers as well

Agree: Additional references to more recent papers (Holden et al., 2012 and Masoud and Koike, 2017) have been included as suggested [Line 2.17]

33 onwards: explain the dimensions 3D versus 2.5D

Agree: The sentence has been rephrased as follows:

*"While this method works well in topography containing slopes <45°, it is inherently limited to 2.5 dimensions (2.5D) as elevation data is gridded in the X-Y plane, causing problems when features crosscut steep or overhanging topography (Pavlis and Mason, 2017)"* [Line 2.34-2.35]

35/36: what do you consider a simple topography? Be more precise.

Agree: "simple topography" has been replaced with "topography containing slopes <45°". [Line 2.34]

2: refrain from using phrases such as "such as described above". Follow this suggestion throughout the manuscript.

Agree: The sentence has been reworded to: *"Unfortunately, the unstructured nature of 3D point data mean that methods for trace detection in raster data, including those previously described, cannot be easily applied."* [Line 2.37-3.38]

4: use the more general term "fractures" instead of "joints"

Agree: "joints" have been replaced with "fractures" as suggested. [Line 6.2]

*Method*

14: "DEM based plane fitting and orientation analysis" didn't work (see general comment III)

Clarify: General comment III is dealt with earlier in this response.

16: To demonstrate the capability of our new computer-assisted trace detection approach,

Agree: The sentence has been rephrased as suggested to avoid using "described above". [Line 4.18]

18: "established manual methods" - A thorough description of the manual method is of great importance as well: Who mapped? At which scale was the mapping performed (see general comment I)?

Agree: Please refer to our response to General Comment I

21: "plutonic diorites and tonalites" - either "plutonic rocks" or "diorites and dolerites" instead

Agree: The sentence has been rephrased as suggested *"… Cretaceous to Paleogene dykes intruding Devonian diorites and tonalities…"* [Line 4.23]

28: use "fractures" instead of "joints" throughout

Disagree: The term "joint" is widely used in the structural geology literature, and conveys useful information (e.g. small displacement, Mode I opening) as opposed to the more generic "fracture". Hence in this specific context (i.e. the small displacement, Mode I fractures observed in the Bingie Bingie and Cape Woolamai datasets) we use the term "joint".

33: "Several sets of systematic and non-systematic joints..." if a joint/fracture is nonsystematic, it doesn't belong to any set.

Agree: "several *sets of joints*" has been replaced with "several *generations of joints*" to avoid this issue. [Line 4.36]

34: "cooling of the intrusion, subsequent deformation and recent unloading." Are these guesses? If not add references, otherwise remove it. What does "recent" mean in geological time scale? Be more precise.

Clarify: These are interpretations based on field-observations and context (and the fact that there are few other processes that generate joint-sets), hence the use of the term "likely". The sentence has been retained (rather than removed as suggested) as the authors feel it provides useful context to the case-study. The timing of unloading is largely poorly constrained (though probably neotectonic), and hence "recent" has been removed.

1: "...accurate orientation measurement" - of what?

Agree: The sentence has been amended to specify that joint-orientation is being measured: *"… allows for accurate joint-orientation measurement …"* [Line 5.5]

*Results*

14: "manual workflows" - the manual extraction method needs to be described for each case study.

Agree: See response to General Comment I

18: "...compare and contrast the results of both manual and assisted interpretations..." The result sections dealing with comparison of the data have to be rephrased in line with newly obtained results (see general comment II). And the closest-point difference calculation needs to be explained in the method section.

**Agree:** We respond to General Comment II earlier in this response. The manuscript has been amended to describe the closest-point difference calculation:

*"Closest-point differences, calculated by subsampling closely-spaced points from each assisted trace and computing the shortest-distance between these and a manually interpreted trace, show that…"*
[Line 6.12 – 5.14]

29: remove "previously mentioned"

**Agree:** The words were unnecessary and have been removed.

31-33: "The results are visually similar..." - see general comment II

**Clarify:** This point is addressed in the response to General Comment II.

1: "As in the previous example,..." Name it.

**Agree:** The example has now been explicitly named: *"As in the Bingie Bingie example, …"* [Line 6.19]

3: What does this difference in numbers of traces reflect? (see also comment to table
1 and general comment I and II)

**Clarify:** The difference in the number of traces interpreted reflects inherent variability of manual interpretation. This point is discussed in detail in our response to General Comment I and III.

8: "as a post processing step..." shorter: "after processing"

**Agree:** Shorter is better!

15: "fault scarps" In the LiDAR DEM (Fig. 4a) the interpreted fault scarps (Figs. 4b and c) are not visible at all. The slope map which was used for doing the interpretation has to be provided in Fig. 4.

**Agree:** The "slope-map" has been included in Fig. 4 as suggested.

27: Replace "described in" by "interpreted by"

**Agree:** "described" has been changed to "interpreted" [Line 7.11]

33 onwards: Here it is unclear whether the errors refer to the 21% pixels located >5 pixels away. "the computed shortest-path ... would "detour" through..." Does this mean that you reduced these errors by adding control points to "guide" the trace? Clarify

**Agree:** The quoted error values refer to unmodified traces. This has been clarified by adding "would" to the sentence: *"A small number of additional control points along these traces would resolve this issue…"* [Line 7.19]

36 "correct fracture zone" A fracture zone cannot be correct or incorrect. Change to for example "desired structural trace"

**Agree:** "Correct" has been replaced with "desired" as suggested. [Line 7.20]

*Discussion*

3: DEM: use plural: DEMs

Agree:  The plural has been used. [Line 7.24]

6: "operates in co-operation" Rephrase

Agree:  The sentences has been rephrased as follows to avoid the repetition:

"… *works in co-operation …*" [Line 7.27]

6/7: change to "user guidance"

Disagree: The authors suggest that "expert-guidance" is an appropriate term here (see response to the previous comment regarding "expert-guidance").

8: if there is a firstly, there has to be somewhere a secondly as well

Agree:  Secondly has been added to the following paragraph ("*Secondly, similarly to*") [Line 7.35]

12: "improved consistency" Yes there is improved consistency, but it doesn't necessarily mean that it increases precision and that it is closer to reality. This remains a bit abstract unless a reference to the true (nature) pattern of the fractures is given.

Agree:  Hence the use of "precision" rather than "accuracy" in line 7.32.

30 "...manually interpret datasets using GeoTrace or Compass..." this is confusing. GeoTrace and Compass are semi-automatic tools. I suggest to use "manual" for the fully-manual extraction method and "semi-automatic" or "computer-assisted" for your new extraction tool consistently throughout the manuscript.

Agree:  The use of "manually" here was incorrect and unnecessary, and so has been removed. The sentence now reads: "*The time it takes for users to interpret datasets using GeoTrace or Compass will vary*" [Line 8.14]

34/35 cf. general comment II

Clarify: Please see our response to General Comment II.

6: add "fully" to "automated"

Agree:  "automated" has been replaced with "fully-automated". [Line 8.29]

Tables and Figures

*Table 1:*
Why do you get different numbers of traces for the manual and the assisted method? Is this due to different operators for each method? Or does this reflect different perception states of the same operator? What does the comparison in numbers tell us here? Cf. general comment II.

**Clarify:** The difference in the number of traces interpreted reflects inherent variability of manual interpretation. This point is discussed in detail in our response to General Comment I and III.

Figures 2-5 should be considered to get redrawn/rearranged according to the general comment II and the comments below.

Figure 2: North arrow is missing in both (a) and (b). Here you could add stereoplots and rosediagrams produced by GeoTrace (cf. general comment III).

**Agree:** A north arrow has been added to Fig. 2a and 2b as suggested.

Figure 3: North arrow is missing in both (a) and (b). What is the reason for the more consistent orientation estimates using the computer-assisted approach (Fig. 3f)? Is this an artefact of the computer-assisted approach? Is the larger spread in orientations in the manual-extracted dataset more realistic? What is closer to reality? See also comment to page 7, line 12. Discuss. caption: get rid of the "similar results" (see general comment II).

**Clarify:** Figure 3a includes a lat/long grid, and hence a north arrow is unnecessary and will only clutter an already busy figure.
**Agree:** A north arrow has been added to 3b and 3d for consistency with 3c.

Figure 4: North arrow is missing. Scale bar: show it from 0 to 1000m only, don't go to -250. What are the straight lines across the DEM? Roads or tractor tracks? Indicate. Show the map you used for extraction of structural traces. In the un-interpreted hillshaded DEM (Fig. 4a) I would not dare drawing any of the interpreted traces. Show the slope map, which was used to extract the traces.

**Clarify:** As above, a north arrow is unnecessary as the figure includes a lat/long grid.
**Agree:** The scale bar has been fixed as suggested. We are unsure what causes all of the "straight lines" across the DEM, though they could quite possibly be farm-tracks or patterns resulting from tractors. Regardless, the lines are not relevant to this publication.

Figure 5:
 Legend for 5a/b (bathymetry?) and for 5c/d (vertical gravity gradient) is missing.

**Agree:** The missing legends have been added to the figure.

The original/uninterpreted data (basemap) needs to be shown for 5a/b. You did the mapping probably at larger scales? If yes, provide zoom-in maps showing exemplary oceanic fracture zone and their interpretation. Indicate location of Fig. 6 in Fig. 5c or 5d.

**Agree:** The original/uninterpreted bathymetry has been shown suggested.  The location of Fig. 6 has been included.

Caption: remove the last sentence (see general comment II)

**Clarify:** This point is discussed in detail in our response to General Comment II.

Add scale and north arrow.

**Disagree:** As in the previous figures, Fig. 5 a-d contain lat/long grids, hence a north arrow is unnecessary and distracting. Also over this extent, a scale-bar would be misrepresentative due to map distortions, hence the use of a lat/long grid.

*Technical corrections:*

1: "sea-stacks" - in other places written without hyphen. Be consistent.

**Agree:** "Sea-stacks" (with a hyphen) is now used consistently throughout the manuscript.

[revised manuscript text omitted]